# Probabilistic Fatigue Life Prediction of Dissimilar Material Weld Using Accelerated Life Method and Neural Network Approach

**Hafiz Waqar Ahmad** [1], **Jeong Ho Hwang** [1], **Kamran Javed** [2], **Umer Masood Chaudry** [3] and **Dong Ho Bae** [1,*]

1   School of mechanical Engineering, Sungkyunkwan University, 2066, Seobu-Ro, Jangan-gu, Suwon, Gyeonggi-do 16419, Korea; waqar543@skku.edu (H.W.A.); reflika@skku.edu (J.H.H.)
2   College of Information and Communication Engineering, Sungkyunkwan University, Suwon 16419, Korea; kamran@skku.edu
3   School of Advanced Materials Science & Engineering, 2066, Seobu-Ro, Jangan-gu, Suwon, Gyeonggi-do 16419, Korea; umer@skku.edu
*   Correspondence: bae@yurim.skku.ac.kr; Tel.: +82-31-290-7443

**Abstract:** Welding alloy 617 with other metals and alloys has been receiving significant attention in the last few years. It is considered to be the benchmark for the development of economical hybrid structures to be used in different engineering applications. The differences in the physical and metallurgical properties of dissimilar materials to be welded usually result in weaker structures. Fatigue failure is one of the most common failure modes of dissimilar material welded structures. In this study, fatigue life prediction of dissimilar material weld was evaluated by the accelerated life method and artificial neural network approach (ANN). The accelerated life testing approach was evaluated for different distributions. Weibull distribution was the most appropriate distribution that fits the fatigue data very well. Acceleration of fatigue life test data was attained with 95% reliability for Weibull distribution. The probability plot verified that accelerating variables at each level were appropriate. Experimental test data and predicted fatigue life were in good agreement with each other. Two training algorithms, Bayesian regularization (BR) and Levenberg–Marquardt (LM), were employed for training ANN. The Bayesian regularization training algorithm exhibited a better performance than the Levenberg–Marquardt algorithm. The results confirmed that the assessment methods are effective for lifetime prediction of dissimilar material welded joints.

**Keywords:** fatigue life prediction; accelerated life testing; Weibull distribution; artificial neural network; bayesian regularization algorithm; dissimilar material weld

## 1. Introduction

Climate change is one of the most difficult challenges facing the world today. To prevent climate change, profound changes in the production, distribution, and consumption of energy are required. The increased emission of carbon dioxide due to various human activities is directly responsible for the increase in the Earth's average temperature. Recently, there has been an immense discussion on environmental protection. To reduce carbon dioxide ($CO_2$) emissions and avoid the related environmental problems, scientists and engineers are always looking for methods with which the emission of exhaust gases can be mitigated. Firstly, the use of renewable energy has become increasingly important in meeting future energy demands and limiting the exposure of $CO_2$, such as solar power plants, wind mills, and geothermal. The key issue associated with greener plants is the amount of energy being extracted from renewable sources of energy, i.e., the energy efficiency of renewable

power plants. For example, the maximum theoretical efficiency of a wind turbine is 40% [1] and that of solar power plants is about 20% [2], while geothermal energy is 12% [3]. There is also an immense amount of ongoing research focused on finding solutions for increasing energy output from renewable energy power plants. Secondly, emission abatement through the improvement of the efficiency of thermal power plants is comparably cost-effective and therefore has great effects on fuel consumption and environmental impact. The development of novel materials for steam turbines and their related components have always been a major issue in the power sector. Steam turbines are exposed to ultra-high temperatures of above 700 °C. During the service period, material is subjected to severe conditions, static and dynamic loads, temperature sequences, weathering, and chemical influences. Nickel alloy 617 has proven to be the most promising due to its high metallurgical stability, oxidation resistance, and ease of fabrication [4]. However, in using this alloy as a structural material for various components, new welding technology is required. Therefore, 12 Cr steel is also a suitable candidate due to its excellent corrosion resistance properties and reasonable cost. The key technology for the application of 12 Cr steel and Alloy 617 to low pressure and temperature stages is to develop and design dissimilar material welding (DMW) technology. Previously, it was suggested that explosion welding (EXW) could be used as a joining method for similar or dissimilar materials [5]. It consists of a solid-state welding process with controlled explosive detonation on the surface of a metal. It was proven that the reflection and superposition of stress waves caused by explosive loading led to redistribution and remarkable reduction of residual welding stresses. ASTM A516 low carbon steel with A5086 aluminum alloy [6,7] and Al–Cu, Ti–Cu, and Cu–Ti [8] explosion bonding have been extensively studied.

In operation, performance deterioration or failure of the critical components can result in huge economic loss or catastrophic consequences, so determining the life time of the key components has now become more important when considering reliability. This prediction is based on how materials behave under stress. As their life time tends to be several years, in this case, an accelerated life test (ALT) becomes a feasible way to accelerate the failure process and shorten the test time. Several ALT models are being used today [9]. In accelerated failure time (AFT) models, it is assumed that failure time will follow the same type of distribution under different levels of stress, and time to failure would be shorter at higher levels of stress [10]. The proportional hazards (PH) model assumes that the applied stresses act multiplicatively on the hazard rate [11]. An extended hazard regression (EHR) model was proposed, which encompasses both the PH and AFT models [12]. Other ALT models also being used are the extended linear hazard regression (ELHR) model [13], proportional mean residual life (PMRL) model [14], and proportional odds (PO) model [15].

Fatigue life prediction can also be done using an artificial neural network, considering the tensile properties, volume fraction, and statistical parameters as the input, and receiving the number of fatigue life cycles as the output. The neural network is supposed to evaluate the degradation on components under mechanical stress in real time to predict when they will eventually fail.

In this work, optimum welding conditions were used to perform dissimilar material welding of Alloy 617 and 12 Cr. The fatigue and corrosion fatigue strength of dissimilar materials welded were found and compared. Fatigue life prediction was done using accelerated life tests and an artificial neural network method. Two training algorithms, Bayesian regularization (BR) and Levenberg–Marquardt (LM), were employed for training ANN. Finally, the effectiveness of the probabilistic prediction methods was investigated.

## 2. Dissimilar Material Welding between Alloy 617 and 12 Cr Steel

Alloy 617 and 12 Cr steel were used as the base materials, while Thyssen 617 was used as the filler material for welding. Alloy 617 and 12 Cr steel were welded together using direct current straight polarity (DCSP) tungsten inert gas (TIG) welding technology. A real-time monitoring system was used to regulate the welding conditions, i.e., heat input, wave form, electrode shape, and the distance between electrodes to workpiece. Dissimilar material welding was performed several times

beforehand by varying different welding parameters and the optimum welding conditions were then adopted for the dissimilar material welding of alloy 617 and 12 Cr steel [16]. Specimens were machined with U-groove for narrow gap welding (Figure 1). The U-groove designed was selected because it is economical and requires less filler metal for welding [17]. That results in less distortion and residual stress-related problems in the dissimilar material weld. Both ends of the base metal plates were fixed using welding jigs to avoid the out-of-plane thermal stresses that might be produced due to the heat input of multipass dissimilar material welding.

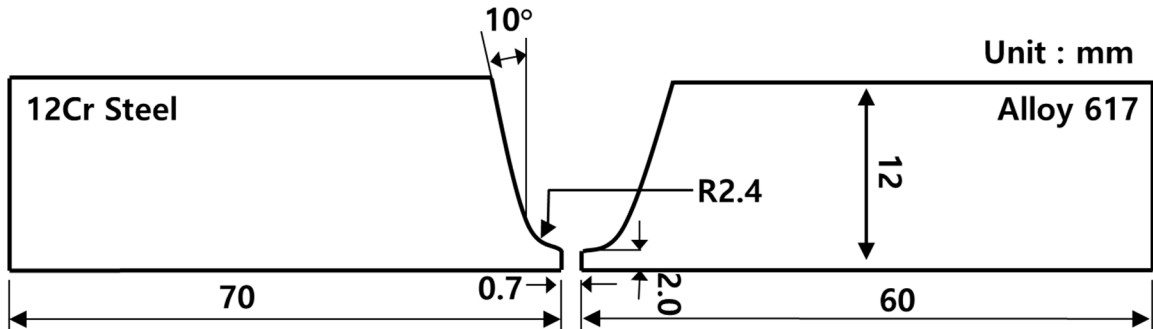

**Figure 1.** Specimen with U-groove for dissimilar material welding.

The composition analysis of alloy 61, 12 Cr steel, Thyssen 617 and dissimilar material weld is given in Table 1. Alloy 617 and Thyssen 617 have relatively similar chemical composition. They both have high nickel content that provides high strength and improved corrosion and oxidation resistance at high temperatures [18]. The magnetic permeability and increased hardenability are due to the cobalt and molybdenum contents in alloy 617. Chromium content is high in 12 Cr steel compared to other elements that considerably increase the hardenability, strength, and response to wear resistance [19]. The composition of dissimilar material weld was quite similar to that of alloy 617 and filler metal, i.e., Thyssen 617.

**Table 1.** Composition analysis of base metals, filler metal, and dissimilar material weld (DMW).

| Base/Filler Metal | Chemical Composition (Weight %) | | | | | | | | | | | |
|---|---|---|---|---|---|---|---|---|---|---|---|---|
| | Ni | Cr | Mo | Co | Al | Fe | C | Si | Mn | Ti | Cu | S |
| Alloy 617 | 44.3 | 22 | 9.0 | 12.5 | 1.2 | 1.5 | 0.07 | 0.5 | 0.5 | 0.3 | 0.2 | 0.008 |
| 12 Cr | 0.43 | 11.6 | 0.04 | - | - | Bal. | 0.13 | 0.4 | 0.58 | - | 0.1 | - |
| Thyssen 617 | 45.7 | 21.5 | 9.0 | 11.0 | 1.0 | 1.0 | 0.05 | 0.1 | - | 1 | - | - |
| DMW | 46.97 | 21.11 | 9.57 | 10.32 | - | 12.03 | - | - | - | - | - | - |

Mechanical properties of base metals, i.e., alloy 617 and 12 Cr steel and their dissimilar material welded joint, are given in Table 2.

**Table 2.** Mechanical properties of Alloy 617, 12 Cr steel, and DMW.

| Material | Yield Strength (MPa) | Tensile Strength (MPa) | Elongation (%) | Reduction in Area (%) | Melting Point (°C) |
|---|---|---|---|---|---|
| **Alloy 617** | 322 | 732 | 62 | 56 | 1330 |
| **12 Cr** | 551 | 758 | 18 | 50 | 1375 |
| **DMW** | 490 | 767 | 48 | - | - |

## 3. Assessing Fatigue Strength of Dissimilar Material Weld

### 3.1. Materials and Test Procedure

The fatigue test specimens were extracted from dissimilar material welded plate as shown in Figure 2. Five specimens were fabricated from dissimilar material welded plates for the assessment of

fatigue strength. All specimens were fabricated in accordance with the ASTM-E8 standard [20]. It can be seen that the gauge length of the specimen consists of the weld metal and HAZ area. The fatigue strength of dissimilar material weld in the air and in a corrosive environment was assessed with a material testing system (INSTRON 8801, 100 kN).

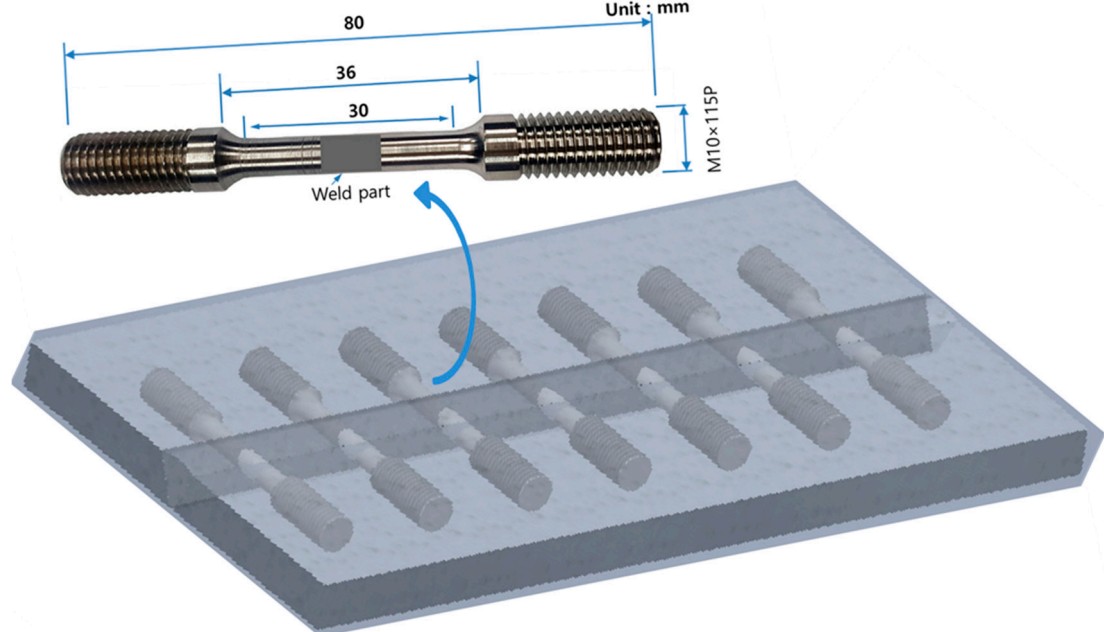

**Figure 2.** Fatigue test specimen configuration.

The loading conditions for the fatigue test in the air and in a corrosive environment are given in Table 3. In short life fatigue tests, the stress levels are suited above the yield stress, and some of the specimens are expected to fail statically at the application of the load. Loading for the fatigue test in the air was started from $\sigma_{max}$ = 690.3, that is, 90% of tensile strength (Table 2) of dissimilar material weld, while for the fatigue test in a corrosion environment, it was started from $\sigma_L$ = 276.12 MPa. A 10% load-decreasing method was used to perform the fatigue test in the air and in a corrosive environment. The load ratio was 0.1. The load frequency for the fatigue test in the air and in a corrosive environment was 10 Hz and 1 Hz, respectively. The lower load frequency for the fatigue test in a corrosive environment is to facilitate the corrosion reaction between the fatigue test specimen and corrosive solution.

**Table 3.** Fatigue test conditions in the air and in a corrosive environment.

| $\sigma_{max}$ = 767MPa (Air) | $\sigma_L$ = 306.8 MPa (Corrosive Environment) | Load Ratio(R) |
|---|---|---|
| $0.9\sigma_u$ = 690.3 | $0.9\sigma_L$ = 276.12 | 0.1 |
| $0.8\sigma_u$ = 613.6 | $0.8\sigma_L$ = 245.44 | |
| $0.7\sigma_u$ = 536.9 | $0.7\sigma_L$ = 214.76 | |
| $0.6\sigma_u$ = 460.2 | $0.6\sigma_L$ = 184.08 | |
| $0.5\sigma_u$ = 383.5 | $0.5\sigma_L$ = 153.4 | |
| $0.4\sigma_u$ = 306.8 | | |

### 3.2. Results and Discussion

Figure 3 shows the S–N curves for the fatigue test specimen in the air and in a corrosive environment. Fatigue tests were carried out on at least five samples in order to assess the variation in values. The variation was negligible. The fatigue limit was assessed as 306.8 and 153.4 in the air and in a corrosive environment [21]. The corrosion fatigue strength of dissimilar material weld specimen was very low compared to its fatigue strength in the air. The presence of a corrosion environment had a

higher influence on the fatigue life of dissimilar material weld [22,23]. The electrochemical dissolution in an aggressive environment reduced the fatigue life of dissimilar material weld [24].

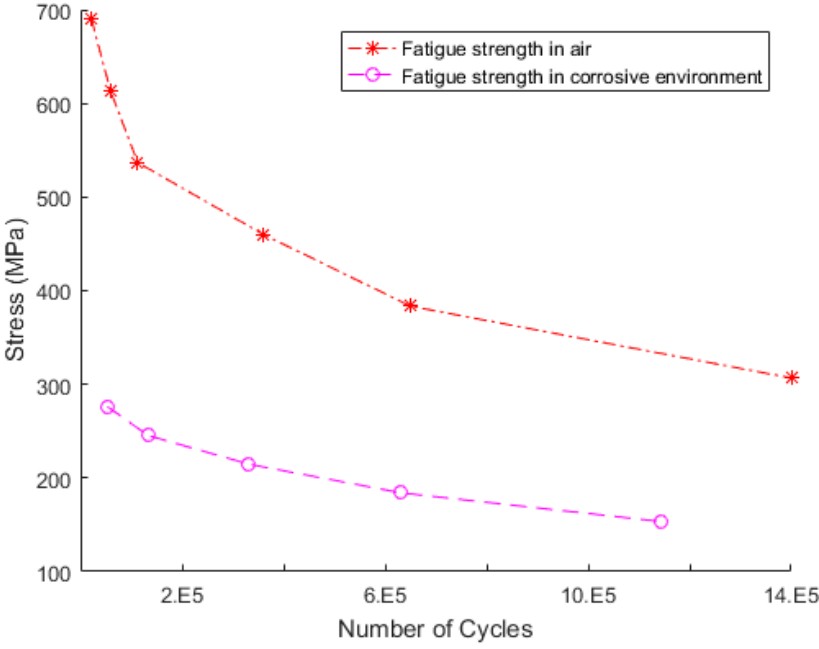

**Figure 3.** Fatigue strength in the air and in a corrosive environment.

## 4. Fatigue Life Prediction Using the Accelerated Life Method

### 4.1. The Goodness-of-Fit Verification

The goodness of fit defines how well fatigue test data fit into a set of observations. The Anderson–Darling (A–D) goodness-of-fit test method deals with the demonstration of a probability distribution. At first, it was found that the distribution corresponds to the fatigue test data, i.e., those acquired from experiments. The basis of confirmation depends on either the A–D statistical value or the *p*-value. The A–D measurement estimates how well the fatigue test data follow a particular distribution. It is the measure of how far the actual data points fall from the fitted line in a probability plot. The statistic is a weighted squared distance from the plot points to the fitted line, and higher weights lie in the tails of the distribution. Generally, a smaller A–D value indicates that the specific distribution better fits the given data.

The Anderson–Darling (A–D) goodness-of-fit test method was implemented using the experimentally-received fatigue test values. A probability plot for different distributions, i.e., normal, log-normal, and exponential and Weibull, were evaluated to find the A–D value for fatigue test data as shown in Figure 4. The A–D value for each distribution is listed in Table 4. It can be seen that in all stress conditions, the Weibull distribution is the best fitted distribution, followed by normal distribution. The Weibull distribution, with the lowest A–D value. is the most appropriate distribution fitted to our data, so this distribution will be further used for accelerated life tests.

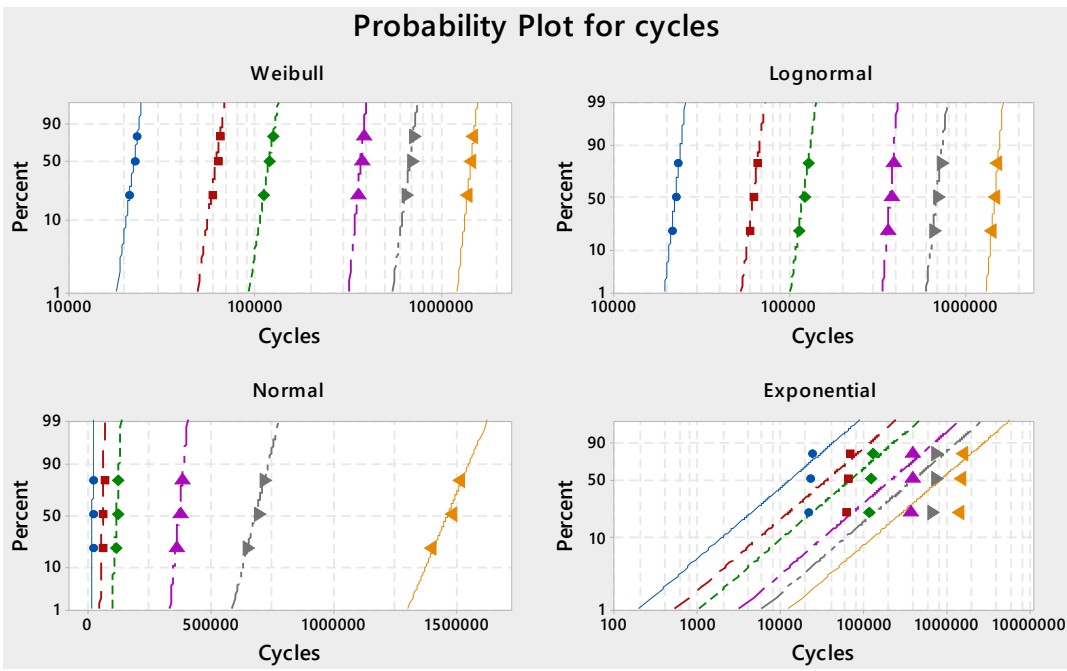

**Figure 4.** Different distributions for Anderson–Darlin (A–D) values.

**Table 4.** Results of Anderson–Darling values for fatigue life.

| Stress Max. (MPa) | Anderson–Darling Value for Different Distributions | | | |
|---|---|---|---|---|
| | **Weibull** | **Log-Normal** | **Normal** | **Exponential** |
| 690.3 | 3.46 | 3.492 | 3.488 | 4.552 |
| 613.6 | 3.446 | 3.468 | 3.464 | 4.539 |
| 536.9 | 3.441 | 3.454 | 3.451 | 4.526 |
| 460.2 | 3.441 | 3.451 | 3.45 | 4.58 |
| 383.5 | 3.442 | 3.458 | 3.455 | 4.548 |
| 306.8 | 3.478 | 3.517 | 3.513 | 4.574 |

*4.2. Accelerated Fatigue Life Verification*

Accelerated life testing (ALT) is more efficient and less costly than traditional reliability testing methods. The goal of ALT is to speed up the failure process to find information about products with a long life well ahead in time. ALT comprises testing under extreme conditions. ALT provides correlation analysis between the fatigue test data at accelerated conditions. This method is used to extrapolate the results back to normal-use conditions. The acceleration between the two given conditions can be established if the regression lines of the accelerated conditions are similar to each other. The regression line slop relates to the shape parameter in the Weibull distribution. The verification method of the acceleration as mentioned above is to validate the resemblance of the slop of regression line that is implemented in the same probability sheet in the Weibull probability distribution. Figure 5 shows the results of the acceleration verification of fatigue test data. The probability plot shows a similar trend without excessive differences. The acceleration of fatigue life test data is considered to be attained as all the regression lines are parallel with 95% reliability for the Weibull distribution. The probability plot for cycles based on the fitted model verifies that accelerating variables at each level was appropriate.

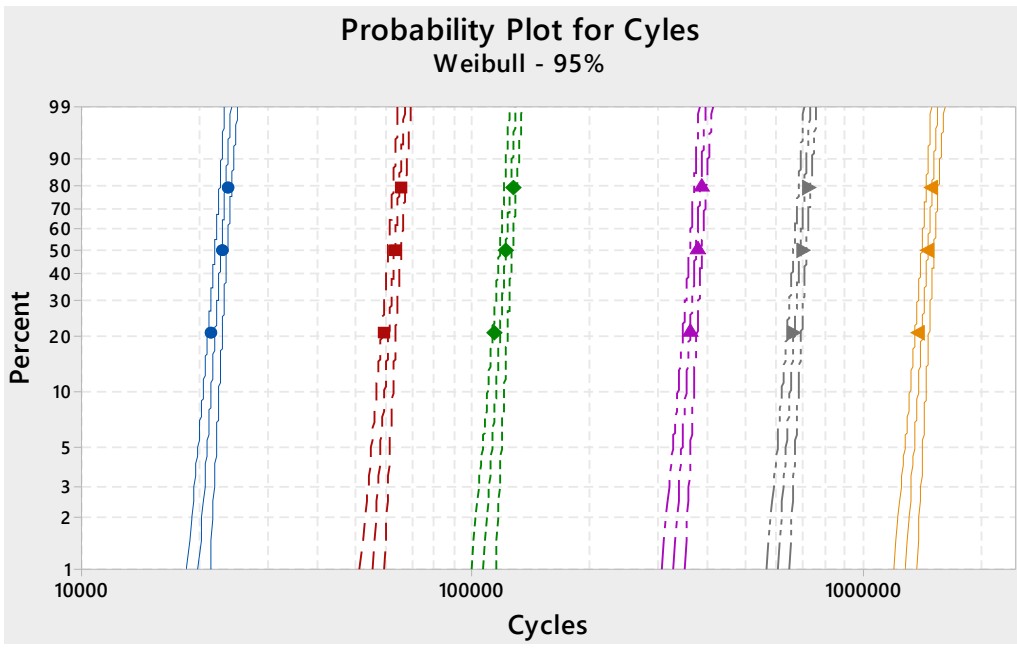

**Figure 5.** Acceleration verification of fatigue life.

The comparison of experimental fatigue life data and accelerated fatigue life prediction is given in Table 5. The accuracy of the fatigue life prediction results was higher than 90%. It can be seen that experimental test data and predicted fatigue life are in good agreement with each other. Therefore, this analysis provides a useful method to predict a specific target life in the region of short lives and high stresses.

**Table 5.** Results of fatigue life prediction.

| No. | Experiment | | Prediction | Accuracy (%) |
|---|---|---|---|---|
| | Stress Max. (MPa) | Fatigue Life (Cycles) | Fatigue Life (Cycles) | |
| 1 | 690.3 | 21,279 | 23,523.83 | 90.5 |
| 2 | 613.6 | 58,846 | 64,778.24 | 90.8 |
| 3 | 536.9 | 112,645 | 123,797.3 | 91.0 |
| 4 | 460.2 | 359,978 | 376,277.9 | 95.7 |
| 5 | 383.5 | 650,000 | 690,486 | 94.1 |
| 6 | 306.8 | 1,400,000 | 1,453,466 | 96.3 |

## 5. Fatigue Life Prediction Using a Neural Network

Neural networks are good at fitting functions. In fact, there is proof that a simple neural network can fit any practical function. A simple neural network to predict fatigue life cycles of material was used. It was trained for welding alloy 617, but the same model can be used for similar materials of same properties.

### 5.1. Artificial Neural Network Architecture

Artificial neural networks (ANNs) [25] consist of multiple artificial neuron AKA perceptrons, as shown in Figure 6, which are used to predict fatigue life cycles of material. Each perceptron has two inputs, tensile strength (MPa) and maximum stress (MPa). Weight $w_{i,j}$ is a real number, expressing the importance of the $i$-th inputs to the $j$-th output. The neuron's output is determined by the weighted sum $\sum_i w_i x_i$, and then it is passed through the activation function [25]. There are

different activation functions, but we chose the sigmoid activation function because it squashes the outputs to the probabilities between 0 and 1, and it is differentiable [26].

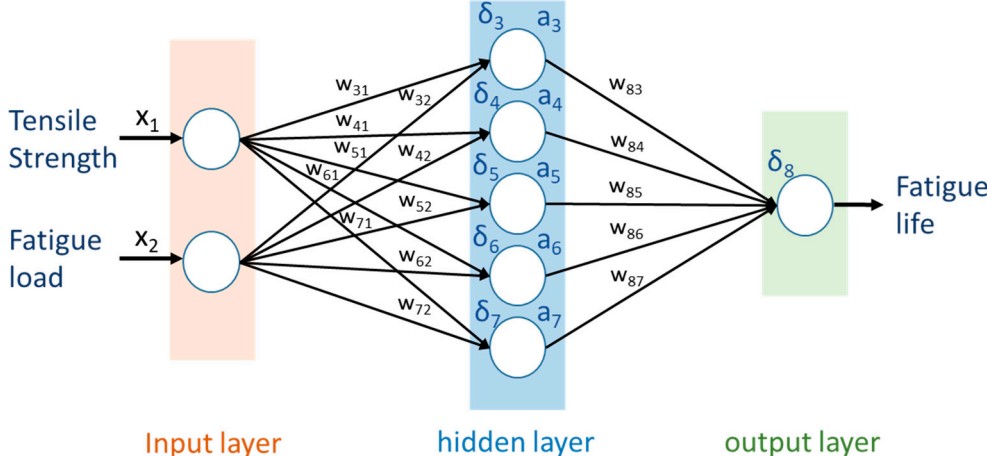

**Figure 6.** Artificial neural network architecture.

ANN consists of three types of neural layers: An input layer, an output, and 10 hidden layers. The weights ($w_{ij}$) and biases are initialized as random values. $\delta_i$ and $a_i$ are the input and activation function of the *i*-th node, respectively. Specifically, we used the sigmoid activation function. The input tensile strength (MPa) value is fed to the input layer of the neural network. Then, the neural network calculates the actual response of each hidden layer and last output layer, which is fatigue life cycles. Finally, the difference between the computed fatigue life cycles and the target fatigue life cycles is calculated, and the connection weights and biases between units are updated to minimize this difference. This iterative update process is repeated for input and target value pairs until the error reduces to a certain value.

*5.2. Dataset for Artificial Neural Network Experiment*

A set of actual experimental tensile strength (MPa) and maximum stress (MPa) values is the inputs of the artificial neural networks. A set of values of fatigue life cycles is the target vectors of the neural networks. Table 6 shows the training data for fatigue life prediction. The training data are divided into three parts: 70% for training, 15% for validation, and 15% for testing.

**Table 6.** Training data for fatigue life prediction.

| Maximum Stress (MPa) | Tensile Strength (MPa) | Life Cycles | Maximum Stress (MPa) | Tensile Strength (MPa) | Life Cycles |
|---|---|---|---|---|---|
| 500.1 | 579.8 | 184,703 | 259.5 | 519 | 502,180 |
| 521.1 | 579.8 | 227,403 | 207.6 | 675 | 701,714 |
| 492.2 | 579.8 | 374,892 | 607.5 | 675 | 41,451 |
| 463.2 | 579.8 | 572,923 | 540 | 675 | 89,208 |
| 434.3 | 579.8 | 639,282 | 472.5 | 675 | 364,802 |
| 405.3 | 579.8 | 792,364 | 405 | 675 | 770,636 |
| 376.4 | 502 | 937,293 | 690.3 | 767 | 21,279 |
| 237.3 | 502 | 38,227 | 613.6 | 767 | 58,846 |
| 211 | 502 | 90,257 | 536.9 | 767 | 112,645 |
| 184.6 | 519 | 321,251 | 460.2 | 767 | 359,978 |
| 415.2 | 519 | 160,140 | 383.5 | 767 | 650,000 |
| 363.3 | 519 | 301,108 | 306.8 | 767 | 1,400,000 |
| 311.4 | 519 | 320,115 | | | |

### 5.3. Training

The convolutional neural network (CNN) model is trained on NVIDIA GPU GTX960. The best performance approach was chosen in which the model reaches its maximum accuracy or lowest square error (MSE). Figure 7 shows the R-Square value [27]; the accuracy of the fitted curve of the trained model is used to predict the fatigue life cycles of material using maximum stress (MPa) only. The overall R-square (R) value is 0.54972.

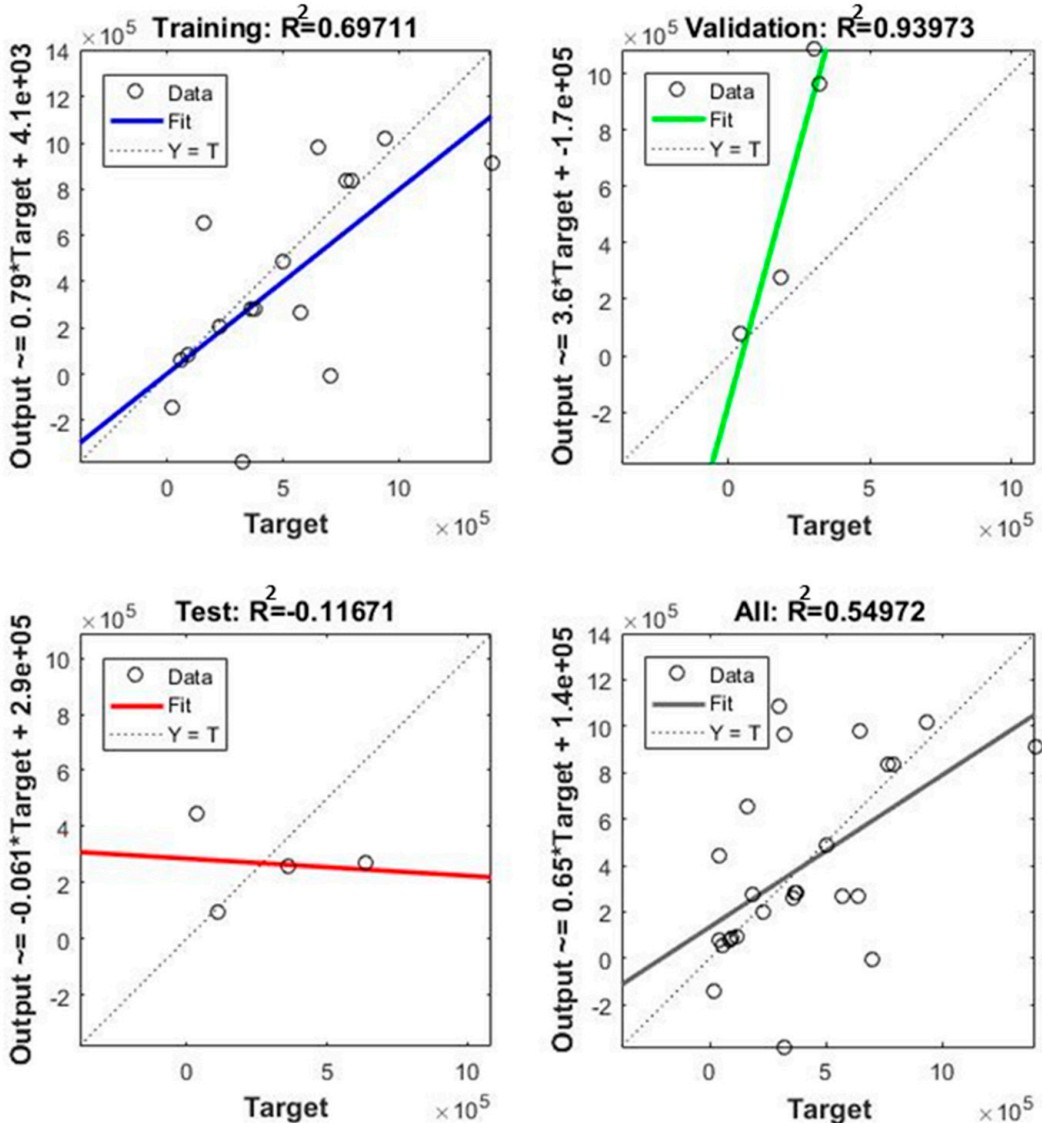

**Figure 7.** Accuracy of fitted curve of trained model.

Figure 8 shows the accuracy of the fitted curve of the trained model to predict fatigue life cycles of material using maximum stress (MPa) and tensile strength (MPa). The overall R-square (R) value is 0.89166, which is better than using maximum stress (MPa) only.

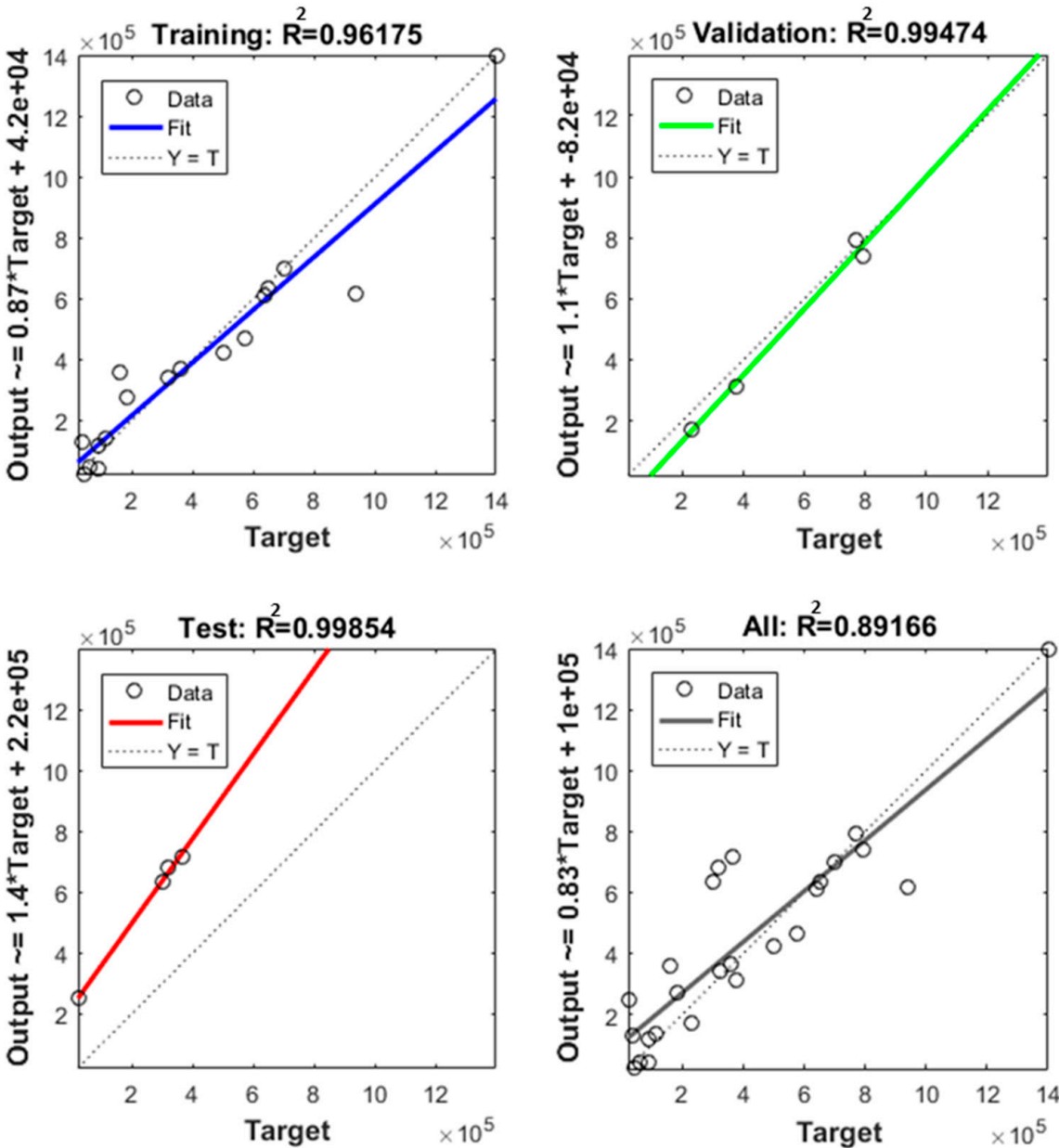

**Figure 8.** Accuracy of fitted curve of trained model.

## 5.4. Fatigue Life Comparison

Predicted fatigue life (cycles) followed the same trend as in the real corrosion fatigue life (cycles), which guaranteed the success of the neural network, as shown in Figure 9. In this section, the corrosion fatigue life predictive ability of artificial neural networks for different training functions was compared. Particularly, two training algorithms, Bayesian regularization (BR) and Levenberg–Marquardt (LM), were used for training ANN [28].

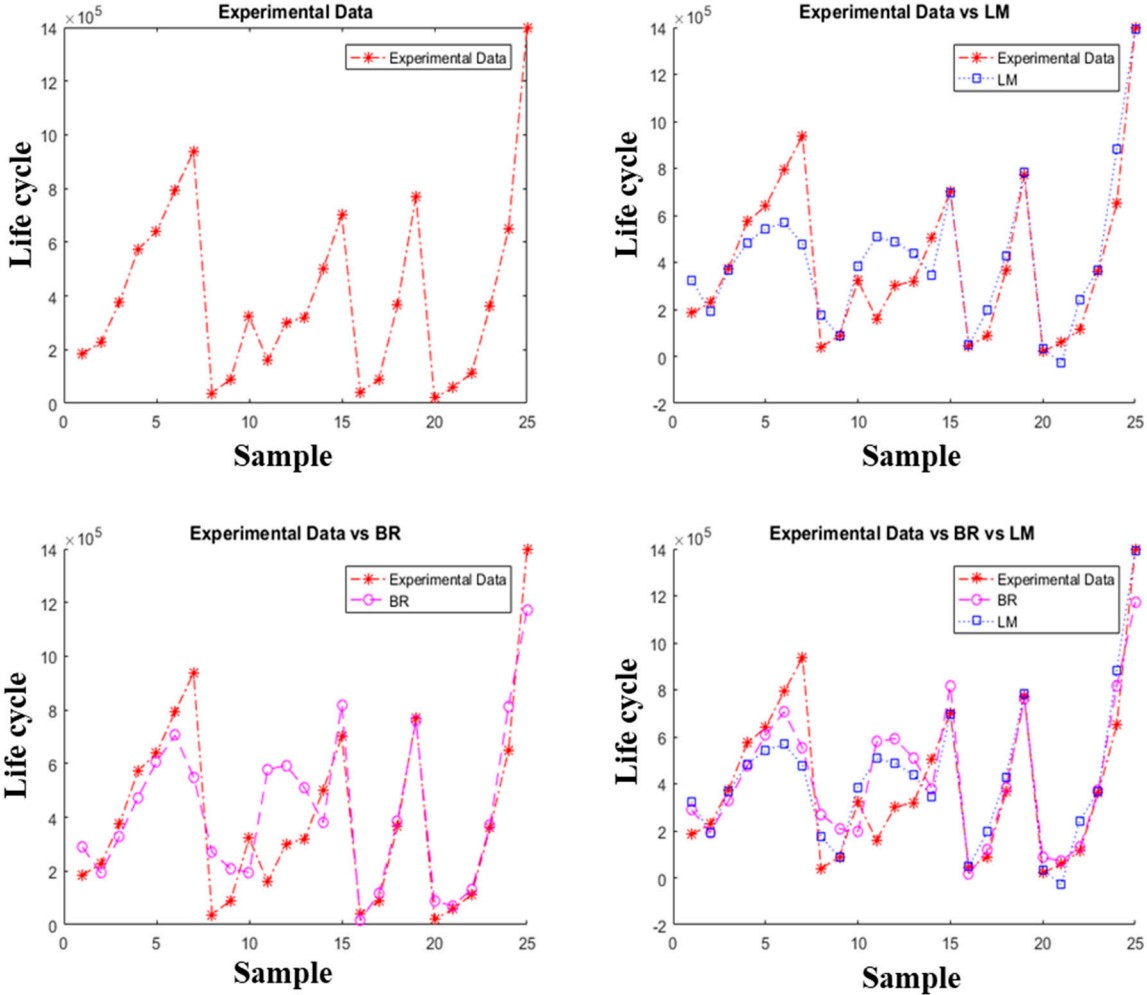

**Figure 9.** Corrosion fatigue life prediction.

It can be seen in Figure 9 that the Bayesian regularization training algorithm has a better performance than the Levenberg–Marquardt algorithm. The advantage of a Bayesian regularization artificial neural network is its ability to reveal potentially complex relationships between inputs and outputs, which means it pledges a more robust model.

## 6. Conclusions

In this paper, the fatigue strength of dissimilar material weld was evaluated in the air and in a corrosive environment. The lifetime assessment of dissimilar weld was predicted through the accelerated life method and artificial neural network approach (ANN), as well. Many conclusions can be drawn based on the results discussed in the previous sections:

1. The fatigue limit of dissimilar material weld was assessed at 306.8 MPa and 153.4 MPa in the air and in a corrosive environment. The electrochemical dissolution in an aggressive environment reduced the fatigue life of dissimilar material weld.
2. The Weibull distribution was found to be the most appropriate distribution that fit the fatigue data well. The acceleration of fatigue life test data was attained with 95% reliability for the Weibull distribution. The accuracy of the fatigue life prediction results was higher than 90%.
3. The corrosion fatigue life of dissimilar material weld predicted by Bayesian regularization (BR) and Levenberg–Marquardt (LM) was in good agreement with the experimentally-obtained results. It seems the Bayesian regularization training algorithm is more accurate, as it can handle the complex relationship between different parameters.

**Author Contributions:** H.W.A. conceived the idea and designed the experiments; J.H.H., H.W.A., and U.M.C. performed the experiments and analyzed the data under the supervision of D.H.B. K.J. did the artificial neural network calculations. H.W.A. and U.M.C. wrote the paper.

**Funding:** This research received no external funding.

**Acknowledgments:** This work was supported by the Reliability Evaluation Laboratory of the Mechanical Engineering Department, Sungkyunkwan University-Suwon, Korea.

**Conflicts of Interest:** The authors have no conflict of interest to disclose.

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
