# Peer review of "Probabilistic Fatigue Life Prediction of Dissimilar Material Weld Using Accelerated Life Method and Neural Network Approach"

_computation, doi:10.3390/computation7010010_

Round 1

Reviewer 1 Report

Please provide microstructure to show how the microstructure is distributed and also tell us where the sample failed on which side did it fail or was it always in the weld. 

Why test above yield strength of the material in both conditions.

Was low cyxle fatigue performed.

The model isgood but was it performed on any other material or on each seperate materials

Author Response

(1) Microstructure

(Reviewer’s Comment)

Please provide microstructure to show how the microstructure is distributed and also tell us where the sample failed on which side did it fail or was it always in the weld. 

(Author’s answer)

Thank you very much for highlighting the very important area. It is pertinent to mention here that authors have already studied in detail, about the microstructure distribution and also have carried out the fractography in their previous study. Below are the details.

[1] H. W. Ahmad, J. H. Hwang, J. H. Lee, D. H. Bae, An assessment of the mechanical properties and microstructure analysis of dissimilar material welded joints between Alloy 617and 12 Cr Steel. Metals 2016, 6(10), 242; https://doi.org/10.3390/met6100242

(2) Conditions for the test

(Reviewer’s Comment)

Why test above yield strength of the material in both conditions.

(Author’s answer)

Depending upon the stress levels, there are various types of fatigue tests i.e Routine, Short-life and long-life. We have done short life test so for short life fatigue test, the stress levels are suited above the yield stress of specimen

(Revised Contents)

“In short life fatigue tests the stress levels are suited above the yield stress and some of the specimens are expected to fail statically at the application of the load.”

(3) Low cycle Fatigue

(Reviewer’s Comment)

Was low cycle fatigue performed?

(Author’s answer)

Thank you very much for highlighting very important aspect. The authors would like to say that our next experimentation is based on the low cycle fatigue in dissimilar material welding.

(4) Model

(Reviewer’s Comment)

The model is good but was it performed on any other material or on each seperate materials.

 (Revised Contents)

“It was trained for welding alloy 617 but the same model can be used for similar materials of same properties”

Reviewer 2 Report

The paper is well organized, It could be considered for publication after the following amendments, which will help increase the quality of the paper:

Dissimilar welding is becoming of outmost importance, the latest technologies arisen and deserve to be mentioned in the introductory part or in section 2, i.e:

[1]          Young GA, Banker JG. Explosion welded, bi-metallic solutions to dissimilar metal joining. 13th Offshore Symposium, Texas Section of the Society of Naval Architects and Marine Engineers, Houston, Texas. 2004.

[2] Corigliano P, Crupi V, Guglielmino E, Sili AM, Full-field analysis of AL/FE explosive welded joints for shipbuilding applications, Mar. Struct. 57 2018; 207-218.

[3] Corigliano P, Crupi V, Guglielmino E. Non linear finite element simulation of explosive welded joints of dissimilar metals for shipbuilding applications, Ocean Eng. 160 2018; 346–353.

[4] Kaya Y. Microstructural, Mechanical and Corrosion Investigations of Ship Steel-Aluminum Bimetal Composites Produced by Explosive Welding, Metals 8 2018, 544.

[5]Findik F. Recent developments in explosive welding, Mater. Des.  32 2011; 1081–1093.

Section 3: How many fatigue tests were carried out for each type of specimen? Was  the dispersion of the results calculated and considered neglectable? What are the fatigue limits observed experimentally?

Line 52: typos,  °C

Line 122 typos: table 2 and not table 3

Section 4. Fatigue life prediction using accelerated life method.

What are A-D values? This section and figure 4 need to be deeper discussed . Is it possible to evaluate an equation for the prediction of  the fatigue life? It shpul be added in the paper.

Figure 7: R-square should be indicated as R2 and not as R only.

Figure 9: please define what you have on x and y axes.

Author Response

Below is the response to reviewer 2.

Round 2

Reviewer 1 Report

Thanks for the revision